# Force transmission through the inner kinetochore is enhanced by centromeric DNA sequences

Elise Miedlar[1†], Grace E Hamilton[1‡], Samuel R Witus[1§], Sara J Gonske[1], Michael Riffle[1#], Alex Zelter[1#], Rachel E Klevit[1], Charles L Asbury[2], Yoana N Dimitrova[3], Trisha N Davis[1*]

[1]Department of Biochemistry, University of Washington, Seattle, United States; [2]Department of Neurobiology and Biophysics, University of Washington, Seattle, United States; [3]Department of Structural Biology, Genentech, Inc, South San Francisco, United States

*For correspondence: tdavis@uw.edu

Present address: [†]University of Pennsylvania, Philadelphia, United States; [‡]Department of Chemistry, High Point University, High Point, United States; [§]Department of Molecular and Cell Biology, University of California, Berkeley, United States; [#]Department of Genome Sciences, University of Washington, Seattle, United States

## eLife Assessment

Centromeres are specific sites on chromosomes that are essential for mitosis and genome fidelity. This **valuable** research advance builds upon previous studies to **convincingly** show that the centromere-histone core contributes to force transduction through the kinetochore. The centromere mainly strengthens one of the two paths of force transduction, influenced by the centromeric DNA sequence. The mechanism underlying this phenomenon will be an exciting future avenue of research, given that centromeric DNAs are not conserved. This work will be of interest to those studying cell division and chromosome segregation.

## Abstract

Previously, we reconstituted a minimal functional kinetochore from recombinant *Saccharomyces cerevisiae* proteins that was capable of transmitting force from dynamic microtubules to nucleosomes containing the centromere-specific histone variant Cse4 (Hamilton et al., 2020). This work revealed two paths of force transmission through the inner kinetochore: through Mif2 and through the Okp1/Ame1 complex (OA). Here, using a chimeric DNA sequence that contains crucial centromere-determining elements of the budding yeast point centromere, we demonstrate that the presence of centromeric DNA sequences in Cse4-containing nucleosomes significantly strengthens OA-mediated linkages. Our findings indicate that centromeric sequences are important for the transmission of microtubule-based forces to the chromosome.

## Introduction

The kinetochore is an assembly of protein subcomplexes that links dynamic microtubule plus ends to chromosomes via centromeric nucleosomes (*Ariyoshi and Fukagawa, 2023*). Through this linkage, kinetochores transmit force from microtubules to chromosomes to align them on the metaphase plate during mitosis, putting both kinetochores and centromeres under tension. This tension signals that kinetochores are correctly bioriented and is a key aspect of the kinetochore's ability to regulate entry into anaphase and prevent aneuploidy (*Akiyoshi et al., 2010*; *Nicklas and Ward, 1994*).

Centromeres in most eukaryotes are defined by the presence of a centromere-specific histone variant, called Cse4 in budding yeast (also known as CENP-A). The yeast 'point' centromere is also defined by DNA sequence – approximately 125 bp in three centromere-determining elements, CDEI, CDEII, and CDEIII (*Clarke and Carbon, 1980*; *Clarke and Carbon, 1985*). Both the Cse4-containing

histone octamer and the centromere-specific DNA elements provide a scaffold for kinetochore assembly. Previously, we showed that kinetochore assemblies built from recombinant *Saccharomyces cerevisiae* proteins can transmit force through either Mif2 or OA as the inner kinetochore subcomplex binding the centromeric nucleosome (*Hamilton et al., 2020*). In that study, we used recombinant nucleosome core particles (NCPs) wrapped with the Widom 601 DNA (W601-NCPs), which has a sequence that wraps canonical histone octamers with high efficiency (*Lowary and Widom, 1998*). The Widom 601 sequence has also fortuitously enabled stable wrapping of centromeric histone octamers in vitro where it would not otherwise be possible. However, Widom 601 DNA does not contain any native centromeric DNA, and therefore we could not address the relationship between centromere-specific DNA and force transmission.

Here, we develop a chimeric DNA sequence (CCEN) that combines native centromeric sequence with Widom 601 sequence to reconstitute centromeric nucleosomes. The chimera includes a portion of CDEII and all of CDEIII from the native centromere to allow for DNA-dependent binding between the NCPs and the kinetochore, as well as segments of Widom 601 DNA at the ends to facilitate stable wrapping. Centromeric histone octamers wrapped with CCEN DNA are stable and thus provide an opportunity to measure the contribution of DNA sequence to the strength of kinetochore attachments.

We found that OA-based assemblies withstood significantly higher forces when nucleosomes were wrapped in the chimeric DNA, while we did not detect significant strengthening of Mif2-based assemblies. Our findings show that centromere-specific DNA sequence elements make significant contributions, either directly or indirectly, to the attachment strength between the inner kinetochore and the centromeric nucleosome.

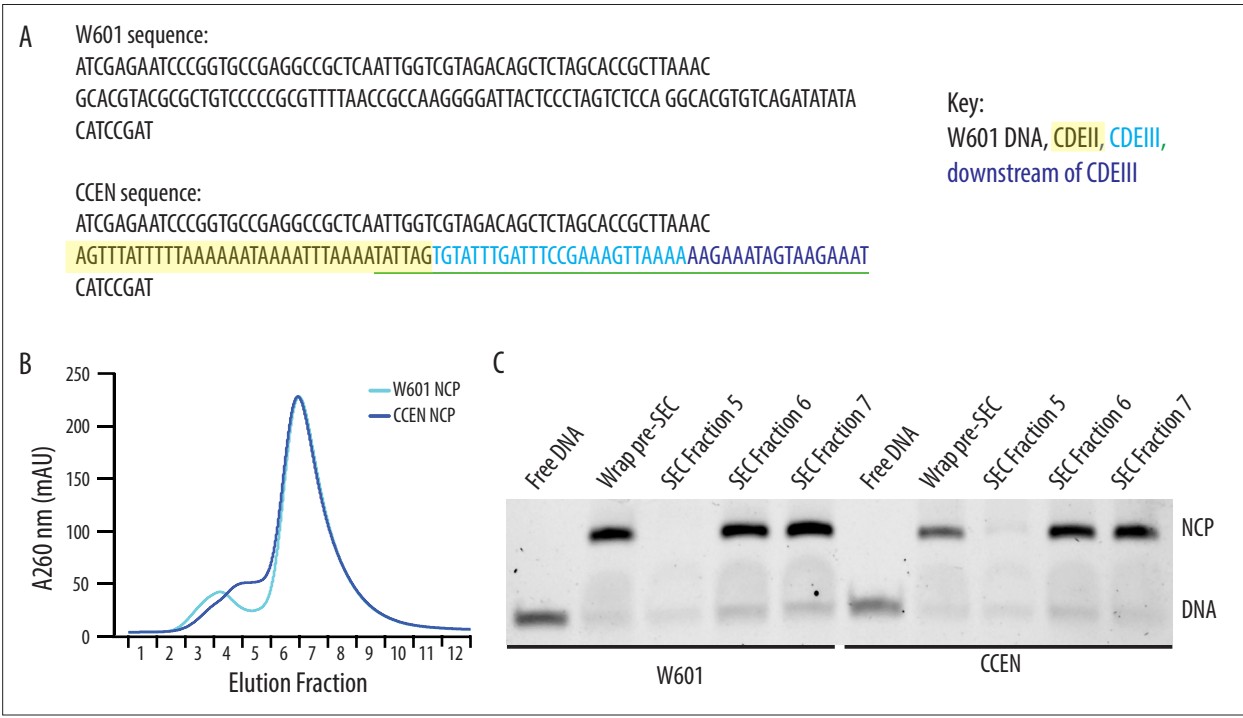

**Figure 1.** Recombinantly wrapped nucleosome core particles (NCPs) with W601 or CCEN DNA. (**A**) Sequences of the W601 and CCEN DNA used to wrap histone octamers. The black text is W601 DNA, the black text highlighted in yellow is the portion of CDEII corresponding to the Mif2 footprint (*Xiao et al., 2017*), the cyan text is CDEIII, the sequence underlined in green is the CBF3 binding site (*Guan et al., 2021*), and the blue text is the pericentromeric DNA sequence just downstream of CDEIII on chromosome III. (**B**) Chromatogram representing elution fractions from size-exclusion chromatography column used to purify wrapped NCPs from excess free DNA. 260 nm signal is shown for both W601 and CCEN NCPs. (**C**) Native gel of elution fractions indicated in the chromatogram in (**B**). NCPs from both SEC Fraction 6 and SEC Fraction 7 were used to collect data on the optical trap. There was no statistically significant difference between rupture forces of assemblies measured with NCPs from either Fraction 6 or Fraction 7.

The online version of this article includes the following source data for figure 1:

**Source data 1.** Native gel for *Figure 1C*, including the relevant bands and conditions.

**Source data 2.** Original file of native gel for *Figure 1C*.

## Results

We produced NCPs by wrapping 147 bp chimeric centromeric DNA (CCEN) or Widom 601 DNA (W601) around recombinant Cse4, H2A, H2B, and H4 histone octamers (*Figure 1A*). The latter three histones had a polyhistidine tag (His$_6$) to facilitate purification and allow NCP binding to the polystyrene beads used in the optical trap rupture force assay. After the initial wrapping, NCPs were purified over a size-exclusion chromatography column (*Figure 1B and C*). As in our previous study (*Hamilton et al., 2020*), we also purified five recombinant kinetochore subcomplexes that can self-assemble onto the NCPs to create a minimal, functional kinetochore: OA, Mif2, MIND, Ndc80c, and Dam1c.

Before determining the quantitative rupture strength that each kinetochore assembly could withstand, we confirmed that the recombinant elements could spontaneously build a functional kinetochore as previously described (*Hamilton et al., 2020*). Briefly, the NCPs bearing His$_6$-tags were bound directly to polystyrene microbeads via anti-His$_6$ antibodies. The remaining subcomplexes, which did not have His$_6$-tags, were added free in solution and bound to the beads only indirectly by assembling with the directly tethered, His$_6$-tagged NCP. Using a laser trap to manipulate individual beads, we tested the kinetochore assemblies for their ability to bind microtubules. Each bead was brought to the dynamic plus end of a microtubule anchored onto the surface of the flow chamber. If the bead bound to the microtubule when released from the laser trap, then the kinetochore assembly was designated as functional, and the bead was used for the rupture force assay.

With either OA or Mif2 in solution, over 78% of beads coated in W601-NCPs or CCEN-NCPs could bind microtubules (*Figure 2A*, *Table 1*). Negative controls where OA or Mif2 was omitted from solution were also performed to confirm the specificity of the NCP for its inner kinetochore binding partner over the outer kinetochore components of the assembly. The negative controls showed 0% bead binding to microtubules, confirming that functional kinetochores are only assembled on NCPs when either OA or Mif2 is present in solution. After a bead was designated as functional, the rupture force of the assembled chain was tested as described in the 'Materials and methods'.

## W601 OA vs. CCEN OA

Although OA has not previously been reported to specifically bind centromeric DNA sequence elements (*Hornung et al., 2014*), kinetochore assemblies that relied on OA were significantly stronger when assembled on CCEN-NCPs relative to those assembled on W601-NCPs (p=1.87 × 10$^{-6}$). The median rupture force for OA-based assemblies on CCEN-NCPs was 6.0 pN compared to the W601-NCP median of 4.2 pN (*Figure 2B and C*).

## W601 Mif2 vs. CCEN Mif2

CCEN-NCP assemblies with Mif2 were not significantly stronger than assemblies with W601-NCPs (p=0.302), which was surprising given that the CDEII region of the centromere that contains a Mif2-binding footprint (*Xiao et al., 2017*) was present in our chimera. The median rupture force for Mif2-based assemblies on CCEN-NCPs was 2.9 pN, which was slightly higher but not significantly different than the W601-NCP median of 2.5 pN (*Figure 2B and C*).

## Discussion

Previous work showed that OA nonspecifically binds to free CEN3 and non-CEN3 DNA fragments, suggesting a sequence-independent OA-DNA binding interaction (*Hornung et al., 2014*). OA is also known to bind the unstructured N-terminus of Cse4 (*Anedchenko et al., 2019*; *Fischböck-Halwachs et al., 2019*; *Shukla et al., 2024*). Here, we find that the strength of OA-based assemblies, when tested in the context of wrapped nucleosomes, is higher when the nucleosomes are wrapped with a CEN3-containing DNA sequence versus when the nucleosomes are wrapped with W601 DNA. This result, together with that of *Hornung et al., 2014*, implies a difference in the importance of centromeric DNA sequences when OA binds to free DNAs as opposed to DNA wrapped around a nucleosome. We propose two possibilities to rationalize this difference. First, the effect could be indirect; an NCP wrapped with the chimeric CCEN DNA might present the N-terminal tail of Cse4 more favorably to OA, thereby increasing the rupture strength of OA-mediated kinetochore assemblies. Alternatively, OA could demonstrate sequence preferences but only when the DNA is bent around a nucleosome and not when presented as free DNA. In support of this possibility, the recent structure of the inner

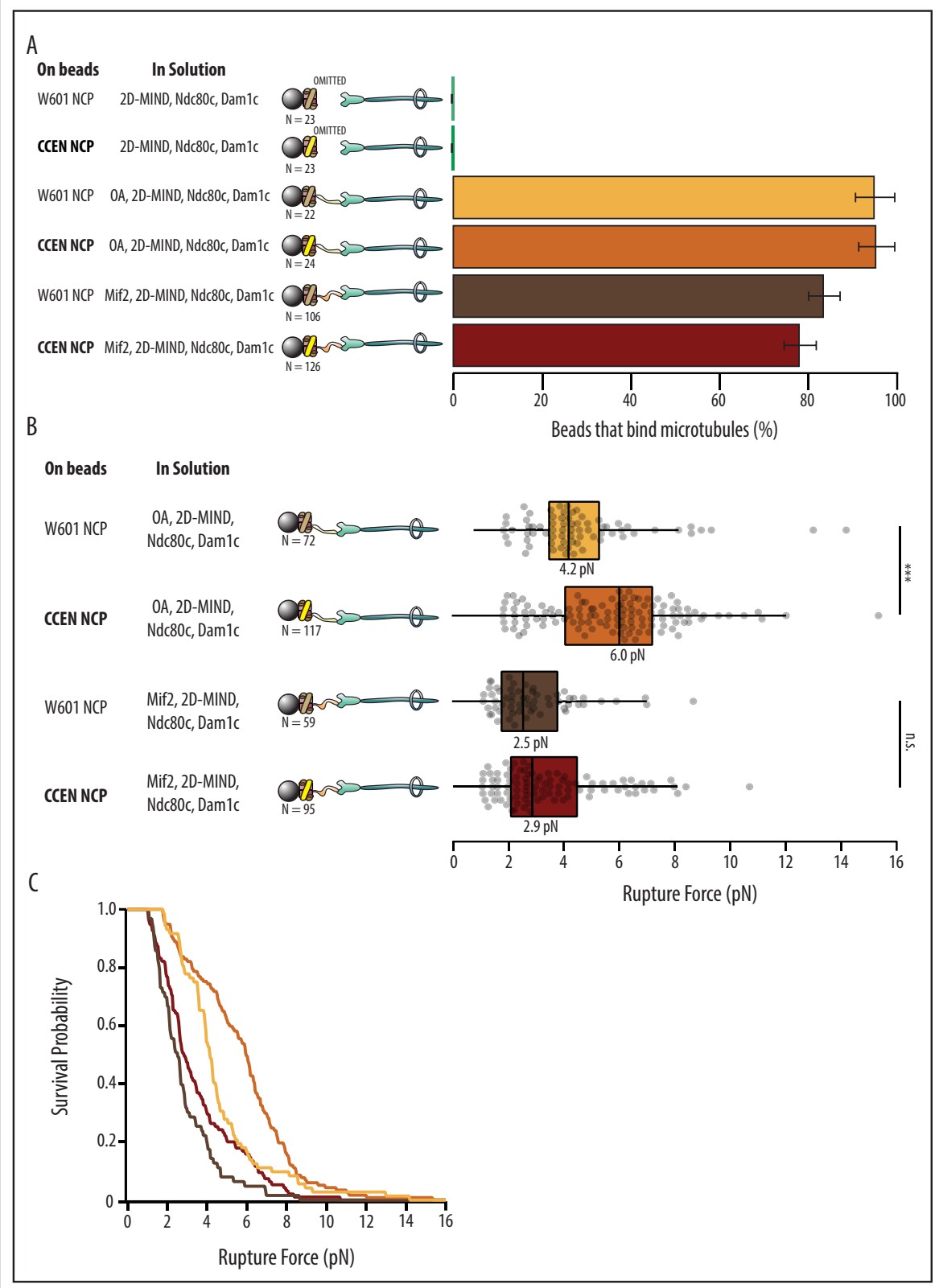

**Figure 2.** Kinetochore assemblies built on CCEN-NCPs and OA form stronger attachments to microtubules. (**A**) Percentages of beads that had microtubule binding capability. Error bars indicate the standard error of the proportion. Barnard's test was used to compare contingency tables. The p-values for the significance of the difference between fraction of beads bound if either OA or Mif2 were included compared to if neither were included are given in *Table 1*. Data were combined from two biological replicates (see *Figure 2—source data 1*). (**B**) Boxplot of rupture forces for each of the

*Figure 2 continued on next page*

*Figure 2 continued*

kinetochore assemblies tested. Dots represent individual rupture events, and boxes enclose the interquartile range, with indicated medians. Whiskers extend to the inner fences. Data were combined from four biological replicates of each condition (see *Figure 2—source data 1*). A Kolmogorov–Smirnov test was performed to compare the probability distributions of rupture forces across conditions. *** indicates p-value of $1.87 \times 10^{-6}$, n.s., not significant. (**C**) Survival probability curves for the data plotted in (**B**).

The online version of this article includes the following source data for figure 2:

**Source data 1.** Excel file of data for *Figure 2A and B* and a list of the number of technical and biological replicates for *Figure 2A and B*.

kinetochore wrapped around a nucleosome shows that a small segment of OA is near the wrapped DNA (*Dendooven et al., 2023*).

*Xiao et al., 2017* showed that both Cse4 and CEN DNA contribute to Mif2 binding to centromeric nucleosomes. Our prior work found that Mif2-based kinetochore assemblies could form on Cse4 containing nucleosomes wrapped with W601 DNA but not on H3-containing nucleosomes wrapped with W601 DNA (*Hamilton et al., 2020*) as expected given the 100-fold lower affinity Mif2 has for H3-based nucleosomes (*Xiao et al., 2017*). Xiao and coworkers also found that Mif2 has a 30-fold higher affinity for CEN3 DNA-wrapped nucleosomes than for nucleosomes wrapped with pericentromeric DNA and 100-fold higher affinity compared to nucleosomes wrapped with W601 DNA. They showed that Mif2 footprints a 35 base pair section at the end of CDEII in CEN3. Here, we wrapped Cse4 nucleosomes with CCEN DNA, which includes all 35 base pairs of the Mif2 footprint in CDEII and compared their properties to those wrapped with W601 DNA. In our experiments, Mif2 did not exhibit significant DNA sequence specificity for CCEN over W601. A key difference between CCEN and CEN3 DNA is that CCEN is missing 47 bp of CDEII. Our results suggest that the increased affinity of Mif2 for CEN3 DNA observed by Xiao and coworkers might require more than just the portion of CDEII that is footprinted by Mif2. Current structures of the yeast inner kinetochore assembled on NCPs either do not include Mif2 or reveal only a peptide of it (*Dendooven et al., 2023*; *Yan et al., 2019*). Our results suggest that a substantial portion of CEN3 might be required to stably assemble Mif2 on the NCP.

It has not yet been possible to obtain structures of yeast NCPs wrapped in authentic CEN3 DNA without using some method to stabilize the NCPs. Guan and coworkers solved a structure of yeast centromeric CEN3 DNA-wrapped NCPs bound to a stabilizing antibody. This structure did not include inner kinetochore proteins (*Guan et al., 2021*). Three structures of yeast NCPs bound to inner kinetochore components have also been solved (*Dendooven et al., 2023*; *Guan et al., 2021*; *Yan et al., 2019*), all of which relied either on W601 DNA or hybrid DNAs that contained 6 bp of CDEII, all of CDEIII, and sequences downstream of CDEIII (Table 3). Here, we describe a new chimera of CEN3 and W601 DNA (CCEN) that includes 35 bp of CDEII as well as all of CDEIII and the region downstream of CDEIII. CCEN stably wrapped yeast centromeric histone octamers and revealed new DNA sequence preferences not previously appreciated.

## Conclusion

We have developed a chimeric DNA sequence (CCEN) that represents an advance toward working with a more native-like recombinant kinetochore. Using centromeric nucleosomes wrapped with CCEN DNA, we have confirmed previous findings that OA and Mif2 can independently support recombinant kinetochore assembly. We showed previously that they can distinguish centromeric versus

**Table 1.** Statistical analyses for microtubule binding assay*.

| Condition | CCEN NCP neither OA nor Mif2 | W601 NCP neither OA nor Mif2 |
|---|---|---|
| W601 NCP, OA | N/A | $2.61 \times 10^{-12}$ |
| CCEN NCP, OA | $3.55 \times 10^{-13}$ | N/A |
| W601 NCP, Mif2 | N/A | $2.31 \times 10^{-12}$ |
| CCEN NCP, Mif2 | $4.27 \times 10^{-7}$ | N/A |

N/A, not applicable.

*p-Values for comparison of data in *Figure 2A* determined by a Barnard's test.

non-centromeric nucleosomes based on histone identity (*Hamilton et al., 2020*). Here, we provide the first evidence that OA also distinguishes between centromeric and non-centromeric nucleosomes on the basis of DNA sequence. We also show that Mif2 requires more than the DNA binding site identified by footprinting to display sequence preferences.

# Materials and methods

## Key resources table

| Reagent type (species) or resource | Designation | Source or reference | Identifiers | Additional information |
|---|---|---|---|---|
| Strain, strain background (*Escherichia coli*) | Rosetta (DE3) pLys competent cells | Novagen | Cat# 71403 | |
| Antibody | MonoRab anti-DYKDDDDK Affinity Resin | GenScript | Cat# L00766 | |
| Biological sample (*Bos taurus*) | Tubulin | Lab purification | | Isolated from *Bos taurus* brains Protocol adopted from *Castoldi and Popov, 2003* |
| Antibody | Biotinylated Anti-His tag antibody | R&D Systems | Cat# BAM050 RRID:AB_356845 | |
| Chemical compound, drug | Glucose oxidase | MilliporeSigma | Cat# 345386 | |
| Chemical compound, drug | Catalase | MilliporeSigma | Cat# E3289 | |
| Chemical compound, drug | Biotinylated bovine serum albumin (BSA) | Vector laboratories | Cat# B-2007 | |
| Chemical compound, drug | Avidin DN | Vector Laboratories | Cat# A-3100 | |
| Chemical compound, drug | TCEP | Thermo Fisher | Cat# 20490 | |
| Other | Streptavidin-coated polystyrene beads | Spherotech | SVP-05-10 | |
| Software | Labview | National Instruments | RRID:SCR_014325 | |
| Software | Igor Pro | Wavemetrics | RRID:SCR_000325 | |

## Plasmids

Plasmids are listed in *Table 2*. Plasmid pAZ144 was derived from pSc_Mf_7, which expressed Mif2-linker-(27-392)MBP-His$_6$ (*Hamilton et al., 2020*). Agilent's QuikChange Lightning Site-Directed Mutagenesis Kit (product number: 210518) was used according to the manufacturer's instructions to loop out the His$_6$ tag from pSc_Mf_7 and modify the C terminal end of the MBP tag to make it identical to the *Escherichia coli* MBP protein. To this end, amino acids GSSHHHHHH were removed and replaced with the native C terminal sequence of the *E. coli* MBP protein (QTRITK), resulting in a new coding

**Table 2.** Plasmids used for expression of kinetochore proteins[*].

| Protein | Plasmid | Components | References |
|---|---|---|---|
| Nucleosome | pScKl2 | *K. lactis* His$_6$-H2A, *K. lactis* His$_6$-H2B, Cse4, *K. lactis* His$_6$-H4 | *Migl et al., 2020* |
| OA | pGH3 | Okp1, Ame1-FLAG | *Hamilton et al., 2020* |
| Mif2 | pAZ144 | Mif2-MBP | This study |
| 2D-MIND | pGH62 | FLAG-Nsl1, S240D & S250D Dsn1, Mtw1, Nnf1 | *Hamilton et al., 2020* |
| (230-576) 2D-MIND | pEHM4 | FLAG-Nsl1, (230-576) S240D & S250D Dsn1, Mtw1, Nnf1 | *Hamilton et al., 2020* |
| Ndc80c | pJT48 Ndc80/Nuf2 | Spc24-FLAG, Spc25 Ndc80, Nuf2 | *Kudalkar et al., 2015*; *Wei et al., 2005* |
| Dam1c | pJT44 | Spc34-FLAG, Dad1, Dad2, Dad3, Dad4 Duo1, Dam1, Hsk3, Spc19, Ask1 | *Umbreit et al., 2014* |

[*]All proteins are from *Saccharomyces cerevisiae* except as noted.

**Table 3.** DNA sequences tested for ability to wrap centromeric nucleosomes.

| Name | Sequence* | Wrapped? | Stable duringSEC? |
|---|---|---|---|
| W601[†] | ATCGAGAATCCCGGTGCCGAGGCCGCTCAATTGGTCGTAGACAGCTCTAGCACCGCTTAAACGCACGTACGCGCTGTCCCCCGCGTTTTAACCGCCAAGGGGATTACTCCCTAGTCTCCAGGCACGTGTCAGATATATACATCCGAT | <u>Yes</u> | Yes |
| CEN 3[‡] | AAAGCTATTCATTGAAAAAATAGTACAAATAAGTCACATGATGATATTTGATTTTATTATATTTTTAAAAAAAGTAAAAAATAAAAAGTAGTTTATTTTTAAAAAATAAAATTTAAAATATTAGTGTATTTGATTTCCGAAAGTTAAAAAAGAAATAGTAAGAAATATATATTT | Some | No |
| CEN 5.1 | AAAGCTATTCATTGAAAAAATAGTACAAATAAGTCACATGATCGAGAATCCCGGTGCCGAGGCCGCTCAATTGGTCGTAGACAGCTCTAGCACCGCTTAAACGCACGTACGCGCTGTCCCCCGCGTTTTAACCGCCAAGGGGATTACTCCCTAGTCTCCAGGCACGTGTCAGATATATACATCCGATTATTTGATTTCCGAAAGTTAAAAAAGAAATAGTAAGAAATATATATTT | Yes | Yes |
| CEN3-601 (*Xiao et al., 2017*) | ATCGAGAATCCCGGTGCCGAGGCCGCTCAATTGGTCGTAGACAGCTCTAGCACCGCTTAAACGCACGTACGCGCTGTCCCCCGCGTTTTAATATTAGTGTATTTGATTTCCGAAAGTTAAAAAAGAAATAGTAAGAAATCATCCGAT | Yes | Yes |
| CCEN[†] | ATCGAGAATCCCGGTGCCGAGGCCGCTCAATTGGTCGTAGACAGCTCTAGCACCGCTTAAACAGTTTATTTTTAAAAAATAAAATTTAAAATATTAGTGTATTTGATTTCCGAAAGTTAAAAAAGAAATAGTAAGAAATCATCCGAT | Yes | Yes |
| CEN 6.3 | ATCGAGAATCCCGGTGCCGAGGCCGCTCAATTGGTCGTAGACAGCTCTAGCACCGCTTAAACAGTTTATTTTTAAAAAATAAAATTTAAAATATTAGAGGGGATTACTCCCTAGTCTCCAGGCACGTGTCAGATATATACATCCGAT | Yes | ND [§] |
| CEN 6.4 | ATCGAGAATCCCGGTGCCGAGGCCGCTCAATTGGTCGTAGACAGCTCTAGCACCGCTTAAACGCACGTACGCGCTGTCCCCCGCGTTTTAACCGCCAAGGGGATTACTCCCTAGTCTCCAGGCACGTGTCAGATATATACATCCGAT**GCGGCC**AGTTTATTTTTAAAAAATAAAATTTAAAATATTAG | Some | ND [§] |
| CON3 [¶] (*Dendooven et al., 2023*) | ATAAGTCACATGGTGCCGAGGCCGCTCAATTGGTCGTAGACAGCTCTAGCACCGCTTAAACGCACGTACGCGCTGTCCCCCGCGTTTTAATATTAGTGTATTTGATTTCCGAAAGTTAAAAAAGAAATAGTAAGAAATATATATTTCATTGAA | ND [§] | ND [§] |

ND, not determined.

*W601 DNA in black font; Genomic DNA upstream or downstream of CEN3 in blue font; CDEI in purple font; CDEII in red font; <u>CDEII</u> Mif2 footprint in red font and underlined; <u>CDEIII</u> in purple font and underlined; **linker DNA** highlighted in bold.

[†]These are the DNA sequences used to wrap the nucleosomes tested in the optical trapping assay in this paper.

[‡]The construct used by *Xiao et al., 2017* included only the 147 bp of CEN3 and none of the upstream or downstream sequences.

[§]Not determined.

[¶]Shown for comparison only.

sequence encoding Mif2-linker-(27-392)MBP. The forward and reverse primers used for this were GAAAGACGCGCAGACTCGTATTACCAAATAATAAACCAACTCCATAAGG and CCTTATGGAGTTGGTTTATTATTTGGTAATACGAGTCTGCGCGTCTTTC, respectively. In all other respects, pAZ144 was identical to pSc_Mf_7. Plasmids are available from the corresponding author.

## Design of chimeric centromeric DNA sequence

The chimeric centromeric DNA sequence (CCEN) is a chimera of CEN3 and Widom 601 DNA. CCEN is 147 bp long and includes four elements: (i) the first 62 bp of W601, (ii) 35 bp of CDEII that includes the Mif2 footprint (*Xiao et al., 2017*), (iii) the rest of the CBF 3 binding site (*Guan et al., 2021*), which is all of CDEIII (25 bp) and 17 bp downstream of CDEIII, and (iv) the last 8 bp of W601. (Note that the 48 bp CBF3 binding site overlaps with the last 6 bp of the Mif2 binding site; *Guan et al., 2021*.) Of the five CEN3-W601 chimeras tested, CCEN was chosen for further study because it included both the Mif2 footprint and the CBF3 binding site, and it stably wrapped centromeric histone octamers such that the

**Table 4.** Kinetochore protein purification buffers.

| Protein | Purification buffers |
|---|---|
| Dam1c-FLAG | **Lysis:** 50 mM sodium phosphate buffer pH 6.9, 500 mM NaCl, 1 mM PMSF, Roche protease inhibitor tablets<br>**FLAG wash:** 50 mM sodium phosphate buffer pH 6.9, 500 mM NaCl, 1 mM PMSF, Roche protease inhibitor tablets<br>**FLAG elution:** 50 mM sodium phosphate buffer pH 6.9, 500 mM NaCl, 1 mM PMSF, Roche protease inhibitor tablets, 200 ug/ml 3x FLAG peptide<br>**SEC:** 50 mM sodium phosphate buffer pH 6.9, 500 mM NaCl |
| Mif2-MBP | **Lysis:** 30 mM HEPES buffer pH 7.5, 2 M NaCl, 10% glycerol, 1 mM TCEP, 1 mM PMSF, Roche protease inhibitor tablets<br>**Amylose resin elution:** 30 mM HEPES buffer pH 7.5, 100 mM NaCl, 10% glycerol, 1 mM TCEP, 1 mM PMSF, Roche protease inhibitor tablets, 10 mM maltose<br>**QA:** 30 mM HEPES buffer pH 7.5, 100 mM NaCl, 10% glycerol, 1 mM TCEP, 1 mM PMSF, Roche protease inhibitor tablets<br>**QB:** 30 mM HEPES buffer pH 7.5, 1 M NaCl, 10% glycerol, 1 mM TCEP |
| MIND-FLAG | **Lysis:** 50 mM HEPES buffer pH 7.5, 200 mM NaCl, 10% glycerol, 0.5% NP40, 1 mM EDTA, 1 mM PMSF, Roche protease inhibitor tablets<br>**FLAG wash:** 50 mM HEPES buffer pH 7.5, 200 mM NaCl, 10% glycerol, 1 mM EDTA, 1 mM PMSF, Roche protease inhibitor tablets<br>**FLAG elution:** 50 mM HEPES buffer pH 7.5, 200 mM NaCl, 10% glycerol, 1 mM EDTA, 1 mM PMSF, Roche protease inhibitor tablets, 200 ug/ml 3x FLAG peptide<br>**SEC:** 50 mM HEPES buffer pH 7.5, 200 mM NaCl, 1 mM EDTA |
| Ndc80c-FLAG | **Lysis:** 50 mM HEPES buffer pH 7.6, 200 mM NaCl, 10% glycerol, 1 mM EDTA, 1 mM PMSF, Roche protease inhibitor tablets<br>**FLAG wash:** 50 mM HEPES buffer pH 7.6, 200 mM NaCl, 10% glycerol, 1 mM EDTA, 1 mM PMSF, Roche protease inhibitor tablets<br>**FLAG elution:** 50 mM HEPES buffer pH 7.6, 200 mM NaCl, 10% glycerol, 1 mM EDTA, 1 mM PMSF, Roche protease inhibitor tablets, 200 ug/ml 3x peptide<br>**SEC:** 50 mM HEPES buffer pH 7.6, 200 mM NaCl, 1 mM EDTA |
| OA-FLAG | **Lysis:** 50 mM HEPES buffer pH 7.5, 200 mM NaCl, 10% glycerol, 0.5% NP40, 1 mM PMSF, Roche protease inhibitor tablets<br>**Low salt FLAG wash:** 50 mM HEPES buffer pH 7.5, 200 mM NaCl, 10% glycerol, 1 mM EDTA, 1 mM PMSF, Roche protease inhibitor tablets<br>**High salt FLAG wash:** 50 mM HEPES buffer pH 7.5, 2 M NaCl, 10% glycerol, 1 mM EDTA, 1 mM PMSF, Roche protease inhibitor tablets<br>**FLAG elution:** 50 mM HEPES buffer pH 7.5, 200 mM NaCl, 10% glycerol, 1 mM EDTA, 1 mM PMSF, Roche protease inhibitor tablets, 200 ug/ml 3x FLAG peptide<br>**SEC:** 50 mM HEPES buffer pH 7.5, 200 mM NaCl, 1 mM EDTA |

nucleosomes could be further purified by size-exclusion chromatography. A construct containing all the CEN3 sequences and no W601 DNA did not stably wrap histone octamers in our hands (*Table 3*).

## NCP wrapping

Yeast histones were purified as described (*Hamilton et al., 2020*). W601 DNA and CCEN DNA were purchased from Integrated DNA Technologies. Mononucleosome core particles were reconstituted by the standard salt dialysis method as described (*Witus et al., 2023*) with two exceptions. First, the nucleosomes contained *S. cerevisiae* histone Cse4, and *K. lactis* histones H2A, H2B, and H4, the latter three were tagged with 6X-His. Second, the W601 DNA and histones were combined in a 1:1.80 molar ratio, while CCEN DNA and histones were combined in a 1:1.55 molar ratio. Nucleosomes were further purified by size-exclusion chromatography using a 24 ml Superdex 200 increase 10/300 column equilibrated in NCP storage buffer (30 mM HEPES buffer [pH 7.5], 10 mM NaCl, 0.1 mM EDTA, 0.5 mM TCEP). NCPs purified by size-exclusion chromatography were stored on ice at 4°C and used within 5 days of purification.

## Purification of kinetochore proteins

OA, Mif2, MIND, Ndc80c, and Dam1c were expressed as described with slight modifications (*Hamilton et al., 2020*). Briefly, each complex was expressed from a polycistronic vector (*Table 2*) in Rosetta 2 DE3 pLysS cells (Novagen). Cells were grown to $OD_{600}$=0.6 and induced with 0.3 mM isopropyl β-D-1-thiogalactopyranoside for 16–18 hours at 18°C. After the induction period, cells were pelleted and washed with PBS containing 1 mM PMSF. Cell pellets were frozen in liquid nitrogen and stored at –80°C until purification.

### MIND-FLAG

On the day of purification, cell pellets were resuspended in MIND-lysis buffer (*Table 4*), and cells were lysed using a French Press and lysates were cleared by high-speed centrifugation. The cleared

lysate was incubated with 2 ml of MonoRab anti-DYKDDDDK Affinity Resin (L00766; GenScript) for 30 minutes. After the resin was washed with 10 column volumes of MIND-wash buffer (*Table 4*), the immobilized protein was incubated for 30 min with MIND-elution buffer (*Table 4*) containing 200 µg/ml 3x FLAG peptide (F4799; Sigma). Eluate from the anti-FLAG resin was collected and further purified by size-exclusion chromatography in MIND-SEC buffer on a 120 ml Superdex 200 column (28-9893-35; GE Healthcare). Eluted fractions were pooled and concentrated if necessary using an Amicon Ultra centrifugal filter (UFC805008; Millipore). Purified protein was stored in MIND-SEC buffer with 5% glycerol at –80°C. Protein concentration was measured via the bicinchoninic acid assay (Thermo Fisher and Sigma-Aldrich).

### OA-FLAG
OA-FLAG was purified as described for MIND-FLAG except that the OA-lysis, OA-elution, and OA-SEC buffers listed in *Table 4* were used. Additionally, the immobilized OA on the FLAG resin was washed with eight-column volumes of OA-high-salt wash and two-column volumes of OA-low-salt wash prior to elution in order to remove co-purifying bacterial DNA.

### Mif2-MBP
Mif2-MBP was purified as described (*Hamilton et al., 2020*), with three exceptions. First, the Mif2-lysis, Mif2-amylose elution, Mif2-QA, and Mif2-QB buffers listed in *Table 4* were used. Second, Mif2 was eluted from the anion exchange column with a 0–80% gradient of QB. Third, Mif2 was not subjected to size-exclusion chromatography.

### Ndc80c-FLAG
Ndc80c-FLAG was purified as described for MIND-FLAG with two exceptions. First, the Ndc80c-lysis, Ndc80c-wash, Ndc80c-elution, and Ndc80c-SEC buffers listed in *Table 4* were used. Second, the eluate from the FLAG affinity resin was concentrated in a 50 kDa molecular weight cutoff concentrator prior to loading on the Superdex 200 column.

### Dam1c-FLAG
Dam1c-FLAG was purified as described for MIND-FLAG, except the Dam1c-lysis, Dam1c-wash, Dam1c-elution, and Dam1c-SEC-specific buffers listed in *Table 4* were used.

## Optical trap assay
Slide preparation for the optical trap assay was performed as previously described (*Flores et al., 2022*; *Hamilton et al., 2020*) with changes to the buffers listed below. 40 nM His6-NCPs and anti-His6 beads were incubated in bead incubation buffer (30 mM HEPES [pH 7.5], 10 mM NaCl, 1 mM EDTA, 0.5 mM TCEP, 2 mg/ml κ-casein) for 30 minutes. All kinetochore proteins were diluted in a buffer containing 1x BRB80 (80 mM PIPES pH 6.9, 1 mM MgCl$_2$, 1 mM EGTA) and 2 mg/ml κ-casein before addition to the reaction mix. The reaction mix contained 1x BRB80, 1 mM GTP, 8 mg/ml BSA, 0.05 mg/ml biotinylated BSA, 0.8 mM DTT, 250 µg/ml glucose oxidase, 30 µg/ml catalase, 3.6 mg/ml glucose, and 16 µM tubulin in addition to the non-His6 tagged proteins free in solution. 10 nM OA or 20 nM Mif2, 10 nM 2D-MIND, 10 nM Ndc80c, and 5 nM Dam1c were used in the trapping assays. Optical trap assays were performed at room temperature using custom instrumentation to capture and manipulate beads as described (*Franck et al., 2010*). Rupture force assays were performed as described (*Hamilton et al., 2020*). Once beads were bound to microtubule tips, a test force of 1 pN was applied, and only beads that tracked with ~100 nm of tip growth were subjected to ramping force of 0.25 pN until detachment. All attachments that withstood the 1 pN preload force were included in our analysis. Each slide was used to collect data for no more than 90 minutes.

## Data analysis and visualization
Igor Pro (Wavemetrics) was used to analyze data from optical trap assays and generate graphs for figures. Adobe Illustrator was used to generate figures.

## Acknowledgements

We thank Mai Bayarjargal, Emmanuel Boakye-Ansah and Eric Muller for helpful discussions. This work was funded by NIH Grants R35GM130293 to TND, R35GM134842 to CLA, and R01CA260834 to REK.

## Additional information

### Competing interests

Yoana N Dimitrova: Affiliated with Genentech, Inc; the author has no other competing interests to declare. The other authors declare that no competing interests exist.

### Funding

| Funder | Grant reference number | Author |
|---|---|---|
| National Institute of General Medical Sciences | R35GM130293 | Trisha N Davis |
| National Institute of General Medical Sciences | R35GM134842 | Charles L Asbury |
| National Cancer Institute | R01CA260834 | Rachel E Klevit |

The funders had no role in study design, data collection and interpretation, or the decision to submit the work for publication.

### Author contributions

Elise Miedlar, Conceptualization, Data curation, Formal analysis, Investigation, Methodology, Writing – original draft; Grace E Hamilton, Conceptualization, Supervision, Investigation, Methodology, Writing – review and editing; Samuel R Witus, Alex Zelter, Supervision, Investigation, Methodology; Sara J Gonske, Investigation, Methodology; Michael Riffle, Data curation, Formal analysis, Validation; Rachel E Klevit, Resources, Supervision, Funding acquisition; Charles L Asbury, Conceptualization, Data curation, Software, Formal analysis, Supervision, Funding acquisition, Investigation, Visualization, Methodology, Project administration, Writing – review and editing; Yoana N Dimitrova, Conceptualization, Resources, Data curation, Formal analysis, Supervision, Funding acquisition, Visualization, Methodology; Trisha N Davis, Conceptualization, Data curation, Formal analysis, Supervision, Funding acquisition, Validation, Investigation, Visualization, Methodology, Project administration, Writing – review and editing

### Author ORCIDs

Elise Miedlar ⓘ https://orcid.org/0009-0009-8209-5210
Grace E Hamilton ⓘ https://orcid.org/0000-0002-0522-0702
Samuel R Witus ⓘ https://orcid.org/0000-0003-1907-8484
Sara J Gonske ⓘ https://orcid.org/0009-0003-5491-9469
Michael Riffle ⓘ https://orcid.org/0000-0003-1633-8607
Alex Zelter ⓘ https://orcid.org/0000-0002-5331-0577
Rachel E Klevit ⓘ https://orcid.org/0000-0002-3476-969X
Charles L Asbury ⓘ https://orcid.org/0000-0002-0143-5394
Trisha N Davis ⓘ https://orcid.org/0000-0003-4797-3152

Reviewer #1 (Public review): https://doi.org/10.7554/eLife.105150.3.sa1
Reviewer #2 (Public review): https://doi.org/10.7554/eLife.105150.3.sa2
Author response https://doi.org/10.7554/eLife.105150.3.sa3

## Additional files

### Supplementary files

MDAR checklist

## Data availability

*Figure 1—source data 2* contains the original image of the native gel shown in Figure 1C. *Figure 2—source data 1* contains the numerical data used to generate Figure 2A and 2B.

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
