## [Editor Report · eLife Assessment]

Centromeres are specific sites on chromosomes that are essential for mitosis and genome fidelity. This **valuable** research advance builds upon previous studies to **convincingly** show that the centromere-histone core contributes to force transduction through the kinetochore. The centromere mainly strengthens one of the two paths of force transduction, influenced by the centromeric DNA sequence. The mechanism underlying this phenomenon will be an exciting future avenue of research, given that centromeric DNAs are not conserved. This work will be of interest to those studying cell division and chromosome segregation.

---

## [Referee Report · Reviewer #1 (Public review)]

Summary:

The authors address the role of the centromere histone core in force transduction by the kinetochore

Strengths:

They use a hybrid DNA sequence that combines CDEII and CDEIII as well as Widom 601 so they can make stable histones for biophysical studies (provided by the Widom sequence) and maintain features of the centromere (CDE II and III).

Weaknesses:

The main results are shown in one figure (Fig 2). Indeed the Centromere core of Widom and CDE II and III contribute to strengthening the binding force for the OA-beads. The data are very nicely done and convincingly demonstrate the point. The weakness is that this is the entire paper. It is certainly of interest to investigators in kinetochore biology, but beyond that the impact is fairly limited in scope.

Comments on revisions:

The additional information provided by the authors will help the reader understand and interpret the manuscript.

---

## [Referee Report · Reviewer #2 (Public review)]

Summary:

This paper provides a valuable addendum to the findings described in Hamilton et al. 2020 (https://doi.org/10.7554/eLife.56582). In the earlier paper, the authors reconstituted the budding yeast centromeric nucleosome together with parts of the budding yeast kinetochore and tested which elements are required and sufficient for force transmission from microtubules to the nucleosome. Although budding yeast centromeres are defined by specific DNA sequences, this earlier paper did not use centromeric DNA but instead the generic Widom 601 DNA. The reason is that it has so far been impossible to stably reconstitute a budding yeast centromeric nucleosome using centromeric DNA.

In this new study, the authors now report that they were able to replace part of the Widom 601 DNA with centromeric DNA from chromosome 3. This makes the assay more closely resemble the in vivo situation. Interestingly, the presence of the centromeric DNA fragment makes one type of minimal kinetochore assembly, but not the other, withstand stronger forces.

Which kinetochore assembly turned out to be affected was somewhat unexpected, and can currently not be reconciled with structural knowledge of the budding yeast centromere/kinetochore. This highlights that, despite recent advances (e.g. Guan et al., 2021; Dendooven et al., 2023), aspects of budding yeast kinetochore architecture and function remain to be understood and that it will be important to dissect the contributions of the centromeric DNA sequence.

In the future, it will be interesting to pinpoint which interactions contribute to the enhanced force resistance in the presence of centromeric DNA.

Strength:

- The paper demonstrates that centromeric DNA can increase the attachment strength between budding yeast microtubules and centromeric nucleosomes.

Weakness:

- How centromeric DNA exerts this effect remains unclear.

Comments on revisions:

I appreciate the authors' detailed response and their decision to list all the tested in chimeras in Table 3.

All my prior comments have been addressed.

---

## [Author Response]

The following is the authors’ response to the original reviews

**Reviewer #1:**
Summary:The authors address the role of the centromere histone core in force transduction by the kinetochore.Strengths:They use a hybrid DNA sequence that combines CDEII and CDEIII as well as Widom 601 so they can make stable histones for biophysical studies (provided by the Widom sequence) and maintain features of the centromere (CDE II and III).Weaknesses:The main results are shown in one figure (Figure 2). Indeed the Centromere core of Widom and CDE II and III contribute to strengthening the binding force for the OA-beads. The data are very nicely done and convincingly demonstrate the point. The weakness is that this is the entire paper. It is certainly of interest to investigators in kinetochore biology, but beyond that, the impact is fairly limited in scope.

This reviewer might have missed that this is a Research Advance, not an article. Research Advances are limited in scope by definition and provide a new development that builds on research reported in a prior paper. They can be of any length. Our Research Advance builds on our prior work, Hamilton et al., 2020 and provides the new result that native centromere sequences strengthen the attachment of the kinetochore to the nucleosome.

**Reviewer #2:**
Summary:This paper provides a valuable addendum to the findings described in Hamilton et al. 2020 (https://doi.org/10.7554/eLife.56582). In the earlier paper, the authors reconstituted the budding yeast centromeric nucleosome together with parts of the budding yeast kinetochore and tested which elements are required and sufficient for force transmission from microtubules to the nucleosome. Although budding yeast centromeres are defined by specific DNA sequences, this earlier paper did not use centromeric DNA but instead the generic Widom 601 DNA. The reason is that it has so far been impossible to stably reconstitute a budding yeast centromeric nucleosome using centromeric DNA.In this new study, the authors now report that they were able to replace part of the Widom 601 DNA with centromeric DNA from chromosome 3. This makes the assay more closely resemble the in vivo situation. Interestingly, the presence of the centromeric DNA fragment makes one type of minimal kinetochore assembly, but not the other, withstand stronger forces.

We thank the reviewer for their careful and positive assessment of our work.

Which kinetochore assembly turned out to be affected was somewhat unexpected, and can currently not be reconciled with structural knowledge of the budding yeast centromere/kinetochore. This highlights that, despite recent advances (e.g. Guan et al., 2021; Dendooven et al., 2023), aspects of budding yeast kinetochore architecture and function remain to be understood and that it will be important to dissect the contributions of the centromeric DNA sequence.

We couldn’t agree more.

Given the unexpected result, the study would become yet more informative if the authors were able to pinpoint which interactions contribute to the enhanced force resistance in the presence of centromeric DNA.Strength:The paper demonstrates that centromeric DNA can increase the attachment strength between budding yeast microtubules and centromeric nucleosomes.Weakness:How centromeric DNA exerts this effect remains unclear.
**Recommendations for the authors:**

**Reviewer #2 (Recommendations for the authors):**
(1) Additional specific mutants would be helpful in interpreting the effect observed. The authors speculate that a small segment of OA near the DNA (based on Dendooven et al., 2023) could be important. Would it be possible to introduce specific mutations and test this?

This would be an interesting study but is far beyond the scope of a Research Advance. In fact, it would make a nice thesis project for a new student. Although perhaps not obvious, these studies require a large set of reagents including wrapped nucleosomes, which must be made fresh (they cannot be frozen) and five purified recombinant complexes, purified by specialized protocols that maintain their activity. Moreover, each datapoint is gathered one at a time. For example, the data in Figure 2 in this manuscript includes 343 datapoints acquired one at a time over the course of 1.5 years.

(2) Please provide the sequences of the other CEN3-W601 chimeras that were tested and did NOT stably wrap centromeric histone octamers. This may help others to design yet different constructs in the future. (Maybe the information is there and I didn't see it?)

We fully agree and thank the reviewer for this excellent suggestion. The sequences and summaries of their wrapping stability are now provided in Table 3, page 17.

(3) I wonder whether the authors tested the C0N3 sequence used in Dendooven et al., 2023. If not, could it be tested? This would more tightly couple the functional assay shown here with the structural work.

We did not test the CON3 sequence, which was published several years after the start of this work. We agree that a tight coupling between the functional assay and the structural work would be useful. However, we also see the advantage of being able to go beyond the structural work and include even more CEN3 sequence than has so far been possible in the structural work.

In addition to measuring the role of DNA sequence in Okp1/Ame1 attachment to the nucleosome, we were interested in the role of DNA sequence in the attachment of Mif2. Therefore, we included all 35 bp of the Mif2 footprint in our chimeric CCEN DNA sequence. CON3 only includes 8 bp from CDEII. We did produce stable nucleosomes using CEN3-601 from Guan et al. (see Table 3). Again, CEN3-601 only includes 8 bp of the Mif2 footprint so we opted to study nucleosomes wrapped in our CCEN DNA with the entire Mif2 footprint. Curiously we found that even the entire Mif2 footprint was not enough to find the DNA sequence specificity seen in the EMSA experiments reported by Xiao et al., 2017.

To help readers understand the differences between all these constructs, we have included them in Table 3.

(4) Would an AlphaFold 3 prediction of the assemblies used in this paper be feasible and useful?

The structures of the Dam1 complex (Jenni et al., 2018), Ndc80 complex (Zahm, et al., 2023 and references therein), MIND complex (Dimitrova et al., 2016), OA complex (Dendooven et al., 2023), and the nucleosome (Xaio et al., 2017; Yan et al., 2019; Guan et al., 2021; Dendooven et al., 2023) are published. The interactions between many of these complexes are understood beyond the level that AlphaFold3 could provide (Dimitrova et al., 2016; Dendooven et al., 2023). One of the main questions is how Mif2 interacts with the nucleosome and the other components of the kinetochore. Even structural analyses that included Mif2 in the assembly detect little or no Mif2 in the final structure. Unfortunately, AlphaFold3 is also not helpful as it predicts only the structure of the dimerization domain, which was already known (Cohen et al., 2008).

AlphaFold3 predicts the rest of Mif2 is largely unstructured with several alpha helices predicted with low confidence.

(5) Given that the centromeric DNA piece included should be able to bind the CBF3 complex, would it be possible to add this complex and test the effect on force transmission?

This would be an interesting experiment, and we do expect CBF3 to bind. As stated above, this is far beyond the scope of this Research Advance. In our experience, with each new kinetochore subcomplex that we add into our reconstitutions, there are new challenges purifying the subcomplex in active form and in sufficient quantity. We are eager to add CBF3 but this is not something we can pull off in the context of this Research Advance. Thank you again for the time and energy spent reviewing our manuscript